# Evaluation of CoronaVac and CoviShield Vaccines on SARS-CoV-2 Infection in Healthcare Workers in Salvador, Brazil

**Jesús Enrique Patiño Escarcina** [1,2,*] ,  **Felipe de Jesus Souza** [1,3] , **Ana Keila Carvalho Vieira Da Silva** [1,3] , **Keile Kemyly Assis Da Silva** [1,3] , **Ruan Barbosa Souza** [1] , **Saulo Reis Nery Santos** [1] and **Eduardo M. Netto** [1]

1   José Silveira Foundation, Center for Research, Learning and Innovation, Salvador 40210-320, BA, Brazil;
    felipe.321.lipe@hotmail.com (F.d.J.S.); anakeila.enf@gmail.com (A.K.C.V.D.S.);
    keilekemylyassis@hotmail.com (K.K.A.D.S.); ruansouzavida@hotmail.com (R.B.S.);
    saulonery_11@hotmail.com (S.R.N.S.); nettoeduardom@hotmail.com (E.M.N.)
2   Collective Health Institute, Federal University of Bahia, Salvador 40110-040, BA, Brazil
3   School of Nursing, Federal University of Bahia, Salvador 40110-040, BA, Brazil
*   Correspondence: gsus.patino@gmail.com; Tel.: +55-(71)-3504-5270

**Abstract:** The emergence and rapid spread of the SARS-CoV-2 Gamma variant in Brazil have raised concerns about SARS-CoV-2 vaccines' neutralizing capacity and viral load impact. Our study aimed to assess the influence of the CoviShield and CoronaVac vaccines on the Ct-N2 value in the healthcare organization's staff who experienced primary SARS-CoV-2 infection. We examined sixty-three COVID-19 cases reported in the first half of 2021 and identified similar clinical and laboratory characteristics among individuals, regardless of the vaccine they received. Surprisingly, our observations revealed that both CoviShield and CoronaVac vaccines had no impact on viral load or the development and severity of symptoms. These findings suggest a potential reduction in neutralizing response and indicate the need to consider the incorporation of other SARS-CoV-2 vaccines and maintain additional containment measures against SARS-CoV-2, as they remain imperative despite vaccination efforts.

**Keywords:** viral load; SARS-CoV-2; vaccines

## 1. Introduction

The Gamma variant (P.1) of Severe Acute Respiratory Syndrome Coronavirus-2 (SARS-CoV-2) first emerged in Brazil in November 2020 [1]. At that time, it raised concern due to its heightened infectivity, contagiousness, and increased risk of severe illness or death compared to prior SARS-CoV-2 variants [1,2]. Throughout 2021, this variant rapidly became the dominant strain in Brazil [3], contributing to over 13 million infections and 350,000 deaths, primarily affecting adult women under 40 years old in the Amazonas state, which became the hardest-hit region and the epicenter of Brazil's health crisis [1,4]. The prevalence of the Gamma variant represented a five-fold increase compared to the rates observed in 2020 when other strains, such as B.1.1.28 and B.1.1.33 of the Beta variant (B.1351), predominated [5,6]. Additionally, it was noted that the Gamma variant had the capability to cause symptomatic infections with an almost two times higher viral load than previous variants [1,7].

In parallel, in January 2021, the National Health Surveillance Agency authorized the emergency use of two COVID-19 vaccines in Brazil, the CoronaVac inactivated adsorbed vaccine (Sinovac) and the viral recombinant vaccine CoviShield (Oxford/AstraZeneca) [8,9]. Both vaccines are administered in two doses, with intervals of 4 and 12 weeks, respectively [10]. During the first half of 2021, 90 million Brazilians (42%, with 14% receiving CoronaVac and 28% receiving CoviShield) received their first vaccine dose, and 70 million

(33%, with 19% receiving CoronaVac and 14% receiving CoviShield) completed their vaccination with two doses [11]. Despite progress in vaccinating the general population, the rapid spread of the Gamma variant raised concerns about its potential to evade immunity, especially in the context of CoronaVac and CoviShield [12], as well as its impact on viral load among infected individuals [13], issues that remained unclear.

The global surveillance of SARS-CoV-2 variants by the World Health Organization (WHO) has provided evidence of the virus's mutation capability [14]. This potential for mutation raises concerns about its ability to evade neutralizing antibodies, even after vaccination [15]. Such mutations have been linked to the emergence of new variants associated with increased rates of COVID-19 transmission. These developments have led to inquiries regarding the effectiveness of vaccines in reducing transmission risk and curbing the spread of various SARS-CoV-2 variants [16]. This study was conducted to evaluate the effect of the CoviShield and CoronaVac vaccines on the Ct value, which serves as an indicator of viral load, and the development and severity of symptoms among healthcare workers with primary SARS-CoV-2 infections.

## 2. Materials and Methods

A cross-sectional study was conducted on data from a larger cohort of healthcare staff associated with the José Silveira Foundation (FJS) units. The FJS is a philanthropic institution with a long-standing presence in the healthcare sector in Salvador, Bahia, Brazil, spanning over 80 years. The FJS operates several specialized units across the city, focusing on diverse health areas, including gynecology and obstetrics, infectious diseases (with a particular emphasis on Tuberculosis), physiotherapy, rehabilitation, chronic pain management, internal medicine, and more. Additionally, the FJS features centralized laboratory facilities, a Specialized Service in Occupational Safety and Medicine (SESMT), and a research, learning, and innovation center. The institution employs various professional categories, encompassing healthcare professionals in specialized fields and administrative personnel. In response to the advent of the COVID-19 pandemic, most of the FJS units remained operational while adapting their work routines. These adaptations aligned with the recommendations and biosecurity measures stipulated by the World Health Organization (WHO), the Brazilian Ministry of Health [17], and internal FJS regulations. The objective was to minimize the transmission risks associated with the healthcare and administrative sectors. The FJS's SESMT established a specialized service to provide immediate healthcare support to its staff members. To this end, all units were instructed to refer individuals displaying flu-like symptoms or those who reported contact with suspected or confirmed COVID-19 cases for clinical evaluation. When necessary, nasopharyngeal swabs were collected to detect the genetic material of SARS-CoV-2. These swabs were obtained on the consultation day by trained laboratory nurses in designated collection rooms using long, flexible swabs to gather samples from the nasopharynx and throat. The collected specimens were then immediately transported to the FJS central laboratory where they were aliquoted and temporarily stored for processing and analysis within the next few hours, utilizing the RT-PCR test (Xpert® Xpress SARS-CoV-2; Cepheid Europe SAS) to identify the virus's genetic material.

The vaccination of FJS workers with CoviShield and CoronaVac was conducted in accordance with the Brazilian National Plan [10] and the Bahia State Vaccination Plan against COVID-19 [18]. This process occurred at various locations throughout Salvador, with appointments scheduled in advance. While workers were free to choose their vaccination site, the type of vaccine administered was determined by vaccine availability at each location, without the intervention of the researchers or the FJS. It's important to note that vaccination was recommended for adults in Brazil but not mandatory. In compliance with the surveillance measures outlined by the regulations of the State of Bahia [18], the SESMT requested that all institution employees provide a certificate or proof of vaccination. This information was systematically collected, organized, and regularly updated.

The data utilized in this analysis are derived from a larger cohort, which comprises staff members from all FJS units who voluntarily agreed to participate by signing a Term of Free and Informed Consent (TCLE). They were subsequently remotely monitored, with intervals of 15 days, during which they reported their vaccination and health statuses. This reporting included information about any flu-like symptoms they encountered, along with details regarding the severity of these symptoms. For the purpose of this study, the dataset specifically included staff members who were either unvaccinated or had received vaccinations with either CoronaVac or CoviShield. The included individuals were monitored between March and August 2021 and declared being diagnosed with SARS-CoV-2 for the first time, information that was confirmed by SESMT clinical records. Those who had been vaccinated with a different vaccine than CoronaVac or CoviShield were excluded from this analysis.

The information incorporated into this study encompassed various worker characteristics, including age, gender, staff category (classified as providing direct patient assistance in healthcare or working in administrative roles), health and vaccination status, clinical features of COVID-19 and their intensity, results of the RT-PCR test, completeness of the immunization schedule (categorized as incomplete with one dose or complete with two doses), and the duration until the detection of SARS-CoV-2 following the last vaccination. The Cycle Threshold (Ct) values for the N2 target, which indicate the number of cycles needed for detecting viral RNA genetic material and serve as a proxy for the SARS-CoV-2 viral load, were directly extracted from laboratory results records of the analyzed samples. Ct values for the N2 target were chosen for analysis over the E target because a single E-target-positive result was considered presumptive positive, and the test required an N2-target-positive result to be categorized as a confirmed positive result [19].

To describe individuals and their clinical characteristics, absolute and relative frequencies were employed for categorical variables, while the median and interquartile range were used for continuous variables. When comparing groups and assessing the effects of vaccination against SARS-CoV-2 among the unvaccinated and those vaccinated with CoronaVac or CoviShield, all the described characteristics underwent statistical univariate comparison based on variable type. Given the relatively small sample sizes observed within the comparison groups, we opted for non-parametric methods, employing Chi-Squared and Kruskal–Wallis tests as appropriate. Due to the limited sample size of each group being compared, it was not feasible to conduct a multivariate analysis to establish the effect size of clinical characteristics on the Ct threshold in relation to vaccination status. All analyses were performed using IBM SPSS v27.0, R software version 4.2.3, and RStudio version 2023.03.0+386.

## 3. Results

Between February and August 2021, the SESMT evaluated 551 workers with flu-like symptoms and subjected them to SARS-CoV-2 detection via RT-PCR testing. Out of this group, 121 tested positive, and 116 of these individuals willingly participated in the large FJS's staff cohort. However, only 63 of them had complete information available and were consequently included in this analysis. At the time of diagnosis, 19.1% ($n = 12$) had not been vaccinated, 36.5% ($n = 23$) had been immunized with CoronaVac, and 44.4% ($n = 28$) with CoviShield. (Table 1)

**Table 1.** Characteristics of health workers with SARS-CoV-2 infection according to the vaccine received (CoronaVac or Covishield) and comparison with unvaccinated workers.

| Characteristics | Total (*n* = 63, 100%) | Unvaccinated (*n* = 12; 19.1%) | CoronaVac (*n* = 23; 36.5%) | CoviShield (*n* = 28; 44.4%) | *p* * |
|---|---|---|---|---|---|
| Female | 46 (73.0%) | 6 (50.0%) | 17 (73.9%) | 23 (82.1%) | 0.11 |
| Age | 42.2 (35.7–50.9) | 34.4 (20.3–48.6) | 45.8 (36.8–52.1) | 41.9 (37.4–47.3) | 0.21 |
| Staff category | | | | | |
| Healthcare | 32 (51.0%) | 3 (25.0%) | 14 (60.9%) | 15 (53.6%) | 0.12 |
| Administrative | 31 (49%) | 9 (75.0%) | 9 (39.1%) | 13 (46.4%) | |
| Comorbidities | 21 (33.3%) | 5 (41.7%) | 6 (26.1%) | 10 (35.7%) | 0.61 |
| Number of doses | | | | | |
| 1 dose | 28 (44.5%) | NA | 9 (39.1%) | 19 (67.9%) | 0.04 [†] |
| 2 doses | 23 (36.5%) | NA | 14 (60.9%) | 9 (32.1%) | |
| Days to diagnosis after the last vaccine dose | 41.0 (19.0–75.0) | NA | 75.0 (21.0–93.0) | 32.0 (18.0–55.5) | 0.01 [†] |
| N2 target Ct value | 19.3 (17.4–22.6) | 20.3 (19.4–24.8) | 18.3 (16.7–22.6) | 19.3 (17.2–21.5) | 0.21 |
| Asymptomatic | 9 (14.3%) | 2 (16.7%) | 5 (21.7%) | 2 (7.1%) | 0.32 |
| Intensity of symptoms | | | | | |
| Mild to moderate | 19 (35.2%) | 5 (50%) | 4 (22.2%) | 10 (38.5%) | 0.30 |
| Severe | 35 (64.6%) | 5 (50%) | 14 (77.5%) | 16 (61.5%) | |

Note: Values are presented as absolute and relative frequencies for categorical variables and median and interquartile range for continuous variables. *: *p*-values correspond to $\chi^2$ and Kruskal–Wallis tests for categorical and continuous variables, respectively. [†]: comparison between vaccinated individuals only. Ct: cycle threshold. NA: not applied.

Among all participants, 73.0% (*n* = 46) were female, and their median age was 42.2 (IQR: 35.7–50.9) years. Concerning occupation, 51.0% (*n* = 32) were healthcare workers directly involved in patient care or exposed to biocontaminated materials, while the remainder held administrative roles. Additionally, 33.3% (*n* = 21) reported having comorbidities. Concerning COVID-19 case characteristics, a minority were asymptomatic (14.3%, *n* = 9), while the majority experienced severe symptoms (64.6%, *n* = 35). In general, the most frequent symptoms of SARS-CoV-2 infection were loss of smell (42.9%; *n* = 27), loss of taste (38.1%; *n* = 24), headache (36.5%; *n* = 23), fatigue and muscle pain (33.3%; *n* = 21), fever 23.8% (*n* = 15) and cough 20.6% (*n* = 13). All these characteristics did not exhibit significant differences (*p* > 0.05) when comparing the unvaccinated group to those who received CoronaVac or CoviShield. (Table 1).

Regarding vaccination status at the time of diagnosis, a greater proportion of participants who received CoronaVac had completed their two-dose regimen (60.9%; *n* = 14), whereas fewer participants immunized with CoviShield had received both doses (32.1%; *n* = 9) (*p* = 0.04). (Table 1).

Regarding the Ct value, there was a similarity between the values for those vaccinated with CoronaVac (med = 18.3; IQR: 16.7–22.6), with CoviShield (med = 19.3; IQR: 17.2–21.5) and the unvaccinated (20.3; IQR: 19.4–24.8) (*p* = 0.21). The time to detect SARS-CoV-2 after the last vaccine dose was longer for those vaccinated with CoronaVac than those immunized with CoviShield (75.0 (IQR: 21.0–93.0) vs. 32.0 (IQR: 18.0–55.5) days; *p* = 0.01) (Table 1 and Figure 1).

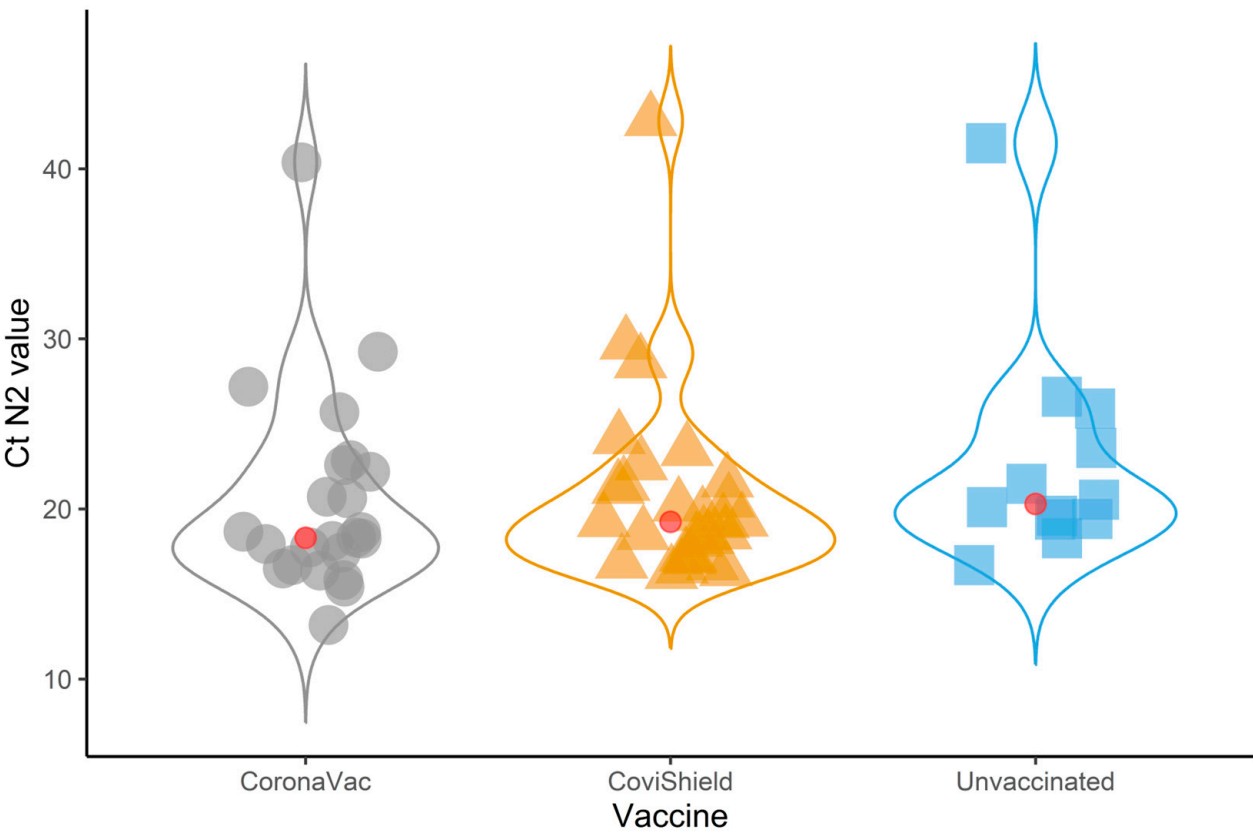

**Figure 1.** Ct N2 values in healthcare workers with SARS-CoV-2 infection according to the vaccine received. (Red dot corresponds to the median Ct N2 value).

## 4. Discussion

In this study, conducted during the prevalence of the SARS-CoV-2 Gamma variant, we observed that the Ct values, which serve as a proxy for viral load in COVID-19 cases, and COVID-19 symptoms were comparable among healthcare workers who were either unvaccinated or had received the CoviShield or CoronaVac vaccines. Regarding the impact of vaccine doses administered, among those who were vaccinated, a higher proportion of COVID-19 cases was observed in individuals who received two doses of CoronaVac compared to those who received two doses of CoviShield. In general, COVID-19 cases were mainly mild or moderate, as reported in other research on the Gamma variant [20], and vaccination appeared to have no impact on clinical characteristics or symptom intensity and was similar for the three groups. Another study finding was the difference in time (days) after the last vaccine dose received and the detection of SARS-CoV-2, with longer protection time for those vaccinated with CoronaVac compared with CoviShield. This discrepancy might be linked to the expected shorter interval between doses for CoronaVac (28 days) compared to CoviShield (90 days) [10]; consequently, a higher percentage of CoronaVac recipients had already received both doses by the time of the study, especially considering that vaccination efforts commenced in January. However, this varying duration of protection should be cautiously explored in a larger sample, considering independent associated factors such as vaccine type, exposure, prior COVID-19 episodes, the SARS-CoV-2 variant, as well as demographic factors like sex, age, and race.

CoronaVac and Covishield vaccines have previously demonstrated their ability to induce an immune response rapidly, intensely, and effectively against SARS-CoV-2. They showed high efficacy against the alpha variant for up to 20 weeks, leading to a significant reduction in deaths, hospitalizations, and severe cases of COVID-19 [16,21–24]. These data underscore the critical role of vaccination for the general population.

The impact of vaccination on viral load has been a subject of investigation due to its potential association with a higher risk of transmission and more severe COVID-19 symptoms [25–27], that if confirmed, could further emphasize the advantages of vaccination. While the effects of vaccination on reducing infections and transmission, possibly due to the reduction in viral load after vaccination [28], were notable during the Alpha variant's dominance [29–32], the vaccine effectiveness may have varied for subsequent variants. This variation was first suggested after studies in animal models which demonstrated reduced efficacy of neutralizing antibodies against the Gamma and Omicron variants compared to Delta variant [33] and was later corroborated in human studies, which found that neutralizing antibody action was lower against Omicron, regardless of vaccination status when compared to the Delta variant [22], which already had Ct values almost five times lower than the Beta variant [34].

Recently, a cohort study has explored viral load dynamics during the prevalence of the Gamma variant under the influence of CoronaVac and Covishield vaccines, suggesting that these vaccines may not significantly decrease SARS-CoV-2 viral load compared to un-vaccinated individuals [21]. Similarly, research on the Delta variant has found even higher SARS-CoV-2 viral loads in unvaccinated individuals compared to those vaccinated with CoronaVac or Covishield [21,22,35–37]. Studies involving other vaccines like Comirnaty (Pfizer-BioNTech) and Spikevax (Moderna) have also failed to identify a significant impact on viral load in Delta variant infections [38]. This evidence supports the hypothesis that the emergence of variants with greater ability to evade neutralizing antibodies is linked to mutations. Therefore, similar to what has been observed with the Delta and Omicron variants [39], the limited effect of vaccination on viral load in Gamma variant infections may have contributed to the rapid spread of the virus despite increased vaccine coverage in Brazil.

Our research did not allow us to rule out the influence of other factors that impact viral load, such as the initial viral load itself, the immune response's characteristics, or the antibody concentration [40]. Nonetheless, the main finding of this study reinforces the hypothesis that vaccination with CoronaVac and Covishield alone may not signifi-cantly reduce the disease's transmission potential; at best, vaccination might have a certain impact on the duration of viral spread, mainly mediated by the associated humoral and cellular immune response. Nonetheless, further research is needed to understand the involved mechanism [15,41–43]. Additionally, this underscores the importance of main-taining supplementary preventive measures alongside vaccination. On the other hand, the evidence may indicate that RNA-based vaccines would better affect the viral load of SARS-CoV-2 variants, reducing the risk of transmission and offering greater benefits to public health [13,44]. Regarding the effect of the number of doses on viral load, few studies have been able to evaluate this hypothesis; however, in infections with the Delta variant, a study was able to identify that a dose of CoviShield was associated with greater protection compared to a dose of CoronaVac [21], which contrasts with our findings.

Our study has certain limitations that need to be acknowledged. Firstly, our relatively small sample size might restrict the generalizability of our findings. Nevertheless, our study shares similarities with research on the Gamma variant and other variants, particularly the Delta variant. Secondly, due to resource constraints, we were unable to conduct genome sequencing surveillance, which would have allowed us to specifically identify Gamma variant infections. It is worth noting that during the study period, approximately 75% of the country genotyped samples analyzed indicated the presence of the Gamma variant [45]. Additionally, we were unable to measure levels of non-neutralizing viral antigen-binding antibodies, which could have provided valuable insights into the immune response against SARS-CoV-2. Lastly, our analysis was based solely on detectable infections, potentially excluding asymptomatic or mild cases with low viral loads that might have been less likely to be identified or reported.

## 5. Conclusions

We can conclude that, while vaccination quickly showcased its benefits by preventing more than 3 million deaths in the region of the Americas [46] and becoming the primary strategy against SARS-CoV-2, it might have a limited impact on the viral load in infections caused by subsequent variants with immune evasion capabilities and higher transmission rates. Furthermore, the vaccination with CoronaVac and CoviShield in Brazil did not suffice to curtail the disease's spread, necessitating the continued implementation of other containment measures.

**Author Contributions:** Conceptualization, J.E.P.E., R.B.S., S.R.N.S. and E.M.N.; methodology, J.E.P.E., F.d.J.S., A.K.C.V.D.S., K.K.A.D.S., R.B.S., S.R.N.S. and E.M.N.; software, J.E.P.E. and E.M.N.; validation, J.E.P.E. and E.M.N.; formal analysis, J.E.P.E. and F.d.J.S.; investigation, J.E.P.E., F.d.J.S., A.K.C.V.D.S., K.K.A.D.S., R.B.S., S.R.N.S. and E.M.N.; data curation, J.E.P.E. and F.d.J.S.; writing—original draft preparation, J.E.P.E., F.d.J.S., A.K.C.V.D.S., K.K.A.D.S., R.B.S. and E.M.N.; writing—review and editing, J.E.P.E., F.d.J.S., K.K.A.D.S. and E.M.N.; visualization, J.E.P.E.; supervision, J.E.P.E.; project administration, E.M.N. All authors have read and agreed to the published version of the manuscript.

**Funding:** This research was entirely self-funded, with all expenses covered by the José Silveira Foundation.

**Institutional Review Board Statement:** The study was conducted in accordance with the Declaration of Helsinki and approved by a Research Ethics Committee of the Climério de Oliveira Maternity (CAE No: 4.712.499 of 14 May 2021).

**Informed Consent Statement:** Informed consent was obtained from all subjects involved in the study.

**Data Availability Statement:** The authors confirm that the data supporting the results and findings of this study are available within the article. The datasets generated during and analyzed during the current study are not publicly available due to privacy but are available from the corresponding author upon reasonable request.

**Conflicts of Interest:** The authors declare no conflict of interest. The funders had no role in the study's design; in the collection, analyses, or interpretation of data; in the writing of the manuscript; or in the decision to publish the results.

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
