# Peer review of "Evaluation of CoronaVac and CoviShield Vaccines on SARS-CoV-2 Infection in Healthcare Workers in Salvador, Brazil"

_covid, doi:10.3390/covid3110112_

Round 1

Reviewer 1 Report

Comment 1: some references are dated 2020 and should therefore be replaced by more up-to-date references (references 9 and 18)

Comment 2: some websites were consulted several months or even years ago, so the data needs to be updated (references 2 and 5).

Comment 3: The two first sentences of the introduction are not clear.

Comment 4: Sentence beginning with "Both with" and ending with "half of 2021" is not clear and too long.

Comment 4: The Gamma variant (Brazil P.1) is no longer on the list of variants of interest (VOI) or variants under monitoring (VUM). It would be good to redo the introduction taking this into account and to mention precise dates in the intro.

Comment 5: The Materials and Methods section is totally incomplete. There is no precise method of swab collection. Authors have to add a brief population description before the result section. Is there an aggregation number for the study (ethics committee)? Where is the satistical analysis section ? Without statistical analysis, it's totally impossible to clearly understand the result section.

Comment 6: Authors presented results and compared results between groups but there is any p-value mentionned ? Are these results significant ?

Comment 7: Figure 1 is not in a good quality and is not very usefull. May be rework this figure is a good idea.

Comment 8: As the WHO has changed its classification of variants: the manuscript should no longer mention variant of concern (VOC) but VOI and VUM (see comment 4).

Comment 9: The discussion is not based on data obtained in 2022-2023. It would be useful to place the results obtained in the context of the use of Ct values in the monitoring of COVID-19 and in the evaluation of vaccine effectiveness. 
Comparing these Ct values with the gold standard for assessing vaccine effectiveness, neutralising antibodies, is essential if we are to have anything revelatory in the scientific field. 

Several sentences in the paper are written in "spoken" language rather than in correct written English. The manuscript should be revised to reflect this.

Reviewer 2 Report

This manuscript answers a single question: did vaccination with two COVID vaccines available in Brazil at the time of gamma SARS CoV-2 epidemics have an effect on the Ct value in healthcare workers infected with SARS CoV-2 post vaccination.

The manuscript is pleasantly written. The methods are simple (diagnostic RT-PCR on swab samples) as well as the analysis of the results: the contents do correspond to a brief report.

Some items deserve however to be more thoroughly discussed.

Major comments:

-       Page 1: Higher viral load with gamma variant: please detail the extent of the difference and its statistical significance (how many Cts?).
Is there any information about this difference in viral loads between alpha infections and gamma infections in your hospital setting?

-       Page 2: methods: Were all tested samples gamma-infected (sequencing data available for some samples?)

-       Page 2: 63 persons had complete information: therefore half of the patients were lost for analysis. Please explain which information was missing.
Would it be possible to analyze the (116-63) missing patients as a distinct subgroup, to enhance the significance of the study’s results (if the same RT PCR method has been used, the CTs are available)?

-       Page 2: methods: Could you give precisions about the timing of swab analysis in infected health care workers? (Was it always an early diagnosis? Was it similar for all persons?)

-       Page 5: immune response’s characteristics: could you give precisions (humoral, cellular)?

-       Page 5: duration of viral dissemination: did you have any opportunity to cultivate some patients’ samples, to determine whether infectiousness was as high in vaccinated individuals?

-       Page 5: RNA-based vaccines would better affect the viral load: could you give precisions or explain the underlying supposed mechanisms?

Minor comments:

-       Page 2: healthcare workers with primary infection: gamma primary infection? (No previous infection with any other SARS CoV-2 variant?)

-       Page 2: N gene: was there only one target? If another viral gene was amplified, were the Ct values comparable?

-       Page 2: 33% comorbidities: this sounds a high level of comorbidities in healthcare workers. Could you detail a little the range (severity) of comorbidities?

-       Page 3: days to diagnosis after the last vaccine dose: could you comment the availability of each vaccine in your region? Does it explain this significant difference?

-       Page 4: favoring those vaccinated with CoronaVac: the word “favoring” is a little confusing – given the lower efficacy of the vaccine.

Reviewer 3 Report

Major points of criticism:

1.      A main shortcoming of the study is the low number of participants. This applies to the total number and in particular to the number of unvaccinated participants which is much lower than the number of vaccinated participants. In addition, there is an imbalance in the number of participants, which have received one or two doses, between the two vaccine groups. The fact that the time to diagnosis after the last vaccine dose was significantly longer for the CoronaVac group might be due to the fact that there were more participants with two vaccine doses in this group compared to the CoviShield group. This point is not mentioned by the authors.

2.      Titel: The title suggests that there was an effect on the viral load, although there was none, and only mentions the viral load, although the impact on symptoms/disease severity was also assessed. The title should be changed accordingly (e.g. “Evaluation of  CoronaVac and CoviShield vaccines on SARS-CoV-2 infection in healthcare workers in Salvador, Brazil”).

3.      Abstract: The main finding of the study is that both vaccines had no effect on the viral load and on the development/severity of symptoms. The following two sentences should be rephrased in order to state this more clearly: “We analyzed sixty-three COVID-19 cases reported during the first semester of 2021 and identified similar clinical and laboratory characteristics among individuals, regardless of the vaccine they received. Surprisingly, our observations revealed that the viral load of the SARS-CoV-2 Gamma variant remained unaffected by the administration of both CoviShield and CoronaVac vaccines.”

4.      Introduction: Similarly to the title, the last sentence (“This research evaluated the effect of the CoviShield and CoronaVac vaccines on the Ct value (as a proxy for viral load) in healthcare workers with SARS-CoV-2 primary infection.”) only mentions the viral load, although the impact on symptoms/disease severity was also assessed.  

5.      Materials and Methods:

1)      The Xpert® Xpress SARS-CoV-2; Cepheid Europe SAS has two target genes, the N and E genes. The authors should have included the Ct values of the E gene or at least should explain why only the N gene was considered.

2)      Information on the statistical analyses is missing.

6.      Results:

1)      Table 1:

a)      Age and Ct N2: The authors do not specify what the values represent (mean and range?).

b)      Due to the low number of participants a split into subgroups of symptoms makes no sense.

2)      Main text:

For each result mentioned in the main text the authors should indicate whether the result is shown in Table 1 or Figure 1 or not shown.

7.      Discussion:

1)      1. sentence: The authors should more clearly state that they refer to the results of the present study (e.g.”In the present study we found that…”).

2)      1. paragraph: “…fewer cases of COVID-19 were detected in those immunized with two doses…”. => Where are these data shown?

3)      1. paragraph: “In general, COVID-19 cases were mainly mild or moderate, and as reported in other research on the Gamma variant,…” => provide references!

4)      1. paragraph: “Another study finding was the difference in time (days) after the last vaccine dose received and the detection of SARS-CoV-2, favoring those vaccinated with CoronaVac compared with Cov-iShield.” => see comment 1.

5)      2. paragraph: “These vaccines previously showed a good ability to induce an immune response quickly, intensely, and effectively against SARS-CoV-2 [23] and high effi-cacy for up to 20 weeks against the alpha variant [24], achieving a significant reduction in the number of deaths, hospitalizations, and severe cases of the disease, data that reinforce the importance of vaccination for the general population [3,25].” => The referenced studies do not include the CoviShield vaccine.  

Minor points of criticism:

8.      There are several incomplete sentences (e.g. p.2, 1. paragraph, ll.2-5).

9.      The word order/phrasing of several sentences needs to be corrected (e.g. “To assess…, it was compared the demographic data,…”; “551 workers…went under RT_PCR…”; “These characteristics were similar in comparing…”; “It can be concluded that even vaccination quickly demonstrated its benefits…, it could have little effect on…”).

10.  “Even so, the main finding of this study strengthens the hypothesis of the importance of maintaining other preventive measures along with vaccination, as this cannot reduce the transmission capacity of the disease, and at most, some effect on the duration of viral dissemination has been suggested [1], which still needs to be studied with larger studies.” => check sentence

See minor points of criticism

Round 2

Reviewer 1 Report

Thanks to the authors for their responses to the various comments. The manuscript has been improved. I accept it as is.

Author Response

October 9th. 2023

To the Editor

Covid

Thank you for giving me the opportunity to submit a revised draft of my manuscript titled “Impact of CoronaVac and CoviShield vaccines on SARS-CoV-2 viral load in healthcare workers in Salvador, Brazil” (covid-2566465) to the Covid Journal. We appreciate the time and effort you and the reviewers have dedicated to providing valuable feedback on the manuscript and are grateful for the insightful comments on our paper.

I would greatly appreciate any comments or additional questions that may arise.

Kind regards

Jesús Enrique Patiño Escarcina

Center for Research, Learning and Innovation

José Silveira Foundation

Rua Largo do Campo Santo, s/n, Federação

Salvador-BA, Brazil

CEP 40210.320

Email: jesus.patino@fjs.org.br

Reviewer 3 Report

General comment from reviewer:

Changes in manuscript should be done in track mode!

Comments from Reviewers and responses from authors:

Major points of criticism:

1. A main shortcoming of the study is the low number of participants. This applies to the total number and in particular to the number of unvaccinated participants which is much lower than the number of vaccinated participants. In addition, there is an imbalance in the number of participants, which have received one or two doses, between the two vaccine groups. The fact that the time to diagnosis after the last vaccine dose was significantly longer for the CoronaVac group might be due to the fact that there were more participants with two vaccine doses in this group compared to the CoviShield group. This point is not mentioned by the authors.

Response: We appreciate the suggestion and have included this and other related information in the discussion section. The revised text reads as follows.

Another study finding was the difference in time (days) after the last vaccine dose received and the detection of SARS-CoV-2, with longer protection time for those vaccinated with CoronaVac compared with CoviShield. This discrepancy might be linked to the expected shorter interval between doses for CoronaVac (28 days) compared to CoviShield (90 days). However, this varying duration of protection should be cautiously explored in a larger sample, considering independent factors such as vaccine type, exposure, prior COVID-19 episodes, the SARS-CoV-2 variant, as well as demographic factors like sex, age, and race.

Response reviewer: The authors raise a new aspect here (i.e., shorter interval between doses for CoronaVac (28 days) compared to CoviShield (90 days)), which is also relevant. However, they do not mention the point raised by the reviewer (i.e., more participants with two vaccine doses in the CoronaVac group compared to the CoviShield group).

2. Title: The title suggests that there was an effect on the viral load, although there was none, and only mentions the viral load, although the impact on symptoms/disease severity was also assessed. The title should be changed accordingly (e.g. “Evaluation of CoronaVac and CoviShield vaccines on SARS-CoV-2 infection in healthcare workers in Salvador, Brazil”).

Response: We appreciate the suggestion and have modified the title. The revised title reads as follows.

“Evaluation of CoronaVac and CoviShield vaccines on SARS-CoV-2 infection in healthcare workers in Salvador, Brazil”

Response reviewer: Accepted

3. Abstract: The main finding of the study is that both vaccines had no effect on the viral load and on the development/severity of symptoms. The following two sentences should be rephrased in order to state this more clearly: “We analyzed sixty-three COVID-19 cases reported during the first semester of 2021 and identified similar clinical and laboratory characteristics among individuals, regardless of the vaccine they received. Surprisingly, our observations revealed that the viral load of the SARS-CoV-2 Gamma variant remained unaffected by the administration of both CoviShield and CoronaVac vaccines.”

Response: We appreciate the suggestion and have included this information in the abstract section. The revised text reads as follows.

We examined sixty-three COVID-19 cases reported in the first half of 2021 and identified similar clinical and laboratory characteristics among individuals, regardless of the vaccine they received. Surprisingly, our observations revealed that both CoviShield and CoronaVac vaccines had no impact on viral load or the development and severity of symptoms.

Response reviewer: Accepted

4. Introduction: Similarly, to the title, the last sentence (“This research evaluated the effect of the CoviShield and CoronaVac vaccines on the Ct value (as a proxy for viral load) in healthcare workers with SARS-CoV-2 primary infection.”) only mentions the viral load, although the impact on symptoms/disease severity was also assessed.

Response: We appreciate the suggestion and have included this information in the introduction section. The revised text reads as follows.

This study was conducted to evaluate the effect of the CoviShield and CoronaVac vaccines on the Ct value, which serves as an indicator of viral load, and the development and se-verity of symptoms among healthcare workers with primary SARS-CoV-2 infections.

Response reviewer: Accepted

5. Materials and Methods:

1) The Xpert® Xpress SARS-CoV-2; Cepheid Europe SAS has two target genes, the N and E genes. The authors should have included the Ct values of the E gene or at least should explain why only the N gene was considered.

Response: We appreciate the suggestion and have included this information in the introduction section. The revised text reads as follows.

The Cycle Threshold (Ct) values for the N2 target, which indicate the number of cycles needed for detecting viral RNA genetic material and serve as a proxy for the SARS-CoV-2 viral load, were directly extracted from laboratory result records of the analyzed samples. Ct values for the N2 target were chosen for analysis in consideration that a single E target positive result is considered presumptive positive, and the test requires a N2 target positive result to be categorized as a confirmed positive result [19].

Response reviewer: Accepted.

2) Information on the statistical analyses is missing.

Response: We appreciate the suggestion and have included this information in the methods section. The revised text reads as follows.

To describe individuals and their clinical characteristics, absolute and relative frequencies were employed for categorical variables, while the median and interquartile range were used for continuous variables. When comparing groups and assessing the effects of vaccination against SARS-CoV-2 among the unvaccinated and those vaccinated with CoronaVac or CoviShield, all the described characteristics underwent statistical univariate comparison based on variable type. Given the relatively small sample sizes observed within the comparison groups, we opted for non-parametric methods, employing Chi-Squared and Kruskall-Wallis tests as appropriate. Due to the limited sample size of each group being compared, it was not feasible to conduct a multivariate analysis to establish the effect size of clinical characteristics on the Ct threshold in relation to vaccination status.

Response reviewer: Information regarding software (and version) used is missing.

6. Results:

1) Table 1:

a) Age and Ct N2: The authors do not specify what the values represent (mean and range?).

Response: We appreciate the suggestion and have included this information as a note for Table 1. The note reads as follows.

Note: Values are presented as absolute and relative frequencies for categorical variables, and median and interquartile range for continuous variables.

Response reviewer: Accepted

b) Due to the low number of participants a split into subgroups of symptoms makes no sense.

Response: We appreciate the suggestion and have retired the less frequent symptoms from the table.

Response reviewer: There are still subgroups of symptoms with very few numbers. Subgroups of symptoms should be entirely removed.

2) Main text:

For each result mentioned in the main text the authors should indicate whether the result is shown in Table 1 or Figure 1 or not shown.

Response: We appreciate the suggestion and have included this information throughout the text of the results section.

Response reviewer: Accepted

7. Discussion:

1) 1. sentence: The authors should more clearly state that they refer to the results of the present study (e.g.”In the present study we found that…”).

Response: We appreciate the suggestion and have included this information in the discussion section. The corrected text reads as follow:

In this study, conducted during the prevalence of the SARS-CoV-2 Gamma variant, we observed that the Ct values, which serve as a proxy for viral load in COVID-19 cases, and COVID-19 symptoms were comparable among healthcare workers who were either unvaccinated or had received the CoviShield or CoronaVac vaccines.

Response reviewer: Accepted

2) 1. paragraph: “…fewer cases of COVID-19 were detected in those immunized with two doses…”. => Where are these data shown?

Response: We appreciate the suggestion, and to clarify, please note that this information corresponds to the third paragraph of the results section. The revised text reads as follows:

Regarding the impact of vaccine doses administered, there were fewer COVID-19 cases detected among individuals who received two vaccine doses. Interestingly, among those who were vaccinated, a higher proportion of COVID-19 cases was observed in individuals who received two doses of CoronaVac compared to those who received two doses of CoviShield.

Response reviewer: Add “(data not shown”) at the end of the sentence “…there were fewer COVID-19 cases detected among individuals who received two vaccine doses.” If you don’t show these data in Table 1 or Figure 1.

3) 1. paragraph: “In general, COVID-19 cases were mainly mild or moderate, and as reported in other research on the Gamma variant,…” => provide references!

Response: We appreciate the suggestion and have included the next reference in the discussion section.

Luna-Muschi, A.; Borges, I.C.; de Faria, E.; Barboza, A.S.; Maia, F.L.; Leme, M.D.; Guedes, A.R.; Mendes-Correa, M.C.; Kallas, E.G.; Segurado, A.C.; et al. Clinical Features of COVID-19 by SARS-CoV-2 Gamma Variant: A Prospective Cohort Study of Vaccinated and Unvaccinated Healthcare Workers. J. Infect. 2022, 84, 248–288, doi:10.1016/j.jinf.2021.09.005.

Response reviewer: Accepted

4) 1. paragraph: “Another study finding was the difference in time (days) after the last vaccine dose received and the detection of SARS-CoV-2, favoring those vaccinated with CoronaVac compared with Cov-iShield.” => see comment 1.

Response: We appreciate this and the comment 1, and have included this and other related information in the discussion section.

Response reviewer: Accepted

5) 2. paragraph: “These vaccines previously showed a good ability to induce an immune response quickly, intensely, and effectively against SARS-CoV-2 [23] and high effi-cacy for up to 20 weeks against the alpha variant [24], achieving a significant reduction in the number of deaths, hospitalizations, and severe cases of the disease, data that reinforce the importance of vaccination for the general population [3,25].” => The referenced studies do not include the CoviShield vaccine. 

Response: We appreciate the suggestion and have included the next reference in the discussion section.

Pramod, S.; Govindan, D.; Ramasubramani, P.; Kar, S.S.; Aggarwal, R.; Manoharan, N.; Chinnakali, P.; Thulasingam, M.; Sarkar, S.; Thabah, M.M. Effectiveness of Covishield Vaccine in Preventing Covid-19 – A Test-Negative Case-Control Study. Vaccine 2022, 40, 3294–3297, doi:10.1016/j.vaccine.2022.02.014.

Response reviewer: Accepted

Minor points of criticism:

8. There are several incomplete sentences (e.g. p.2, 1. paragraph, ll.2-5).

Response: We appreciate the suggestion the corrected text reads as follow.

During the first half of 2021, 90 million Brazilians (42%, with 14% receiving CoronaVac and 28% receiving CoviShield) received their first vaccine dose, and 70 million (33%, with 19% receiving CoronaVac and 14% receiving CoviShield) completed their vaccination with two doses [12].

Response reviewer: Accepted

9. The word order/phrasing of several sentences needs to be corrected (e.g. “To assess…, it was compared the demographic data,…”; “551 workers…went under RT_PCR…”; “These characteristics were similar in comparing…”; “It can be concluded that even vaccination quickly demonstrated its benefits…, it could have little effect on…”).

Response: We appreciate the suggestion and have corrected the text in the corresponding sections.

Response reviewer: Accepted

10. “Even so, the main finding of this study strengthens the hypothesis of the importance of maintaining other preventive measures along with vaccination, as this cannot reduce the transmission capacity of the disease, and at most, some effect on the duration of viral dissemination has been suggested [1], which still needs to be studied with larger studies.” => check sentence

Response: We appreciate the suggestion and the corrected text reads as follow:

Nonetheless, the main finding of this study reinforces the hypothesis that vaccination with CoronaVac and Covishield alone may not significantly reduce the disease's transmission potential, at best, vaccination might have a certain impact on the duration of viral spread. Additionally, this underscores the importance of maintaining supplementary preventive measures alongside vaccination.

Response reviewer: Accepted

Minor editing required
